# ADAR1p150 Forms a Complex with Dicer to Promote miRNA-222 Activity and Regulate PTEN Expression in CVB3-Induced Viral Myocarditis

**DOI:** 10.3390/ijms20020407

**Published:** 2019-01-18

**Authors:** Xincai Zhang, Xiangting Gao, Jun Hu, Yuxin Xie, Yuanyi Zuo, Hongfei Xu, Shaohua Zhu

**Affiliations:** 1Institute of Forensic Medicine, Soochow University, Suzhou 215021, China; xczhang1@stu.suda.edu.cn (X.Z.); 11916@jsmc.edu.cn (J.H.); yxxie@stu.suda.edu.cn (Y.X.); snxie@stu.suda.edu.cn (Y.Z.); 2Department of Pathology, School of Medicine, Shihezi University, Shihezi 215021, China; 20154221095@stu.suda.edu.cn

**Keywords:** ADAR1p150, viral myocarditis, miRNA-222, PTEN

## Abstract

Adenosine deaminases acting on RNA (ADAR) are enzymes that regulate RNA metabolism through post-transcriptional mechanisms. ADAR1 is involved in a variety of pathological conditions including inflammation, cancer, and the host defense against viral infections. However, the role of ADAR1p150 in vascular disease remains unclear. In this study, we examined the expression of ADAR1p150 and its role in viral myocarditis (VMC) in a mouse model. VMC mouse cardiomyocytes showed significantly higher expression of ADAR1p150 compared to the control samples. Coimmunoprecipitation verified that ADAR1p150 forms a complex with Dicer in VMC. miRNA-222, which is involved in many cardiac diseases, is highly expressed in cardiomyocytes in VMC. In addition, the expression of miRNA-222 was promoted by ADAR1p150/Dicer. Among the target genes of miRNA-222, the expression of phosphatase-and-tensin (PTEN) protein was significantly reduced in VMC. By using a bioinformatics tool, we found a potential binding site of miRNA-222 on the PTEN gene’s 3′-UTR, suggesting that miRNA-222 might play a regulatory role. In cultured cells, miR-222 suppressed PTEN expression. Our findings suggest that ADAR1p150 plays a key role in complexing with Dicer and promoting the expression of miRNA-222, the latter of which suppresses the expression of the target gene PTEN during VMC. Our work reveals a previously unknown role of ADAR1p150 in gene expression in VMC.

## 1. Introduction

It has been well documented that viral myocarditis (VMC) is a primary cause of sudden death and heart failure in young adults [1,2,3]. Coxsackievirus B3 (CVB3), belonging to the Picornaviridae family of genus Enterovirus [4], is regarded as the most common infectious pathogen responsible for VMC and the subsequent pathogenesis of dilated cardiomyopathy (DCM) [5]. CVB3 induces VMC either through inflammatory mechanisms or through a direct cytopathic effect (CPE) [6,7,8,9]. The prevalence of acute myocarditis identified through routine necropsy is significantly higher than that through clinical diagnosis [10,11], and the possible explanation for this discrepancy is that only a certain percentage of individuals with acute myocarditis develop easily observable clinical manifestations. Consistent with this, another study identified acute myocarditis as the cause of death at a rate of ~1.5% in a total of 2560 autopsies but was suspected in only one of the cases in the clinic [12]. This indicates that >97% of patients with lethal myocarditis do not have clinical presentations. Symptomless myocarditis may induce severe electrical dysfunction in the heart, and may even be life-threatening, particularly during exertion, which may explain the sudden death of young and vigorous individuals [13,14]. Therefore, CVB-induced VMC is not an unusual disease. Since it has a potentially lethal outcome, it is imperative to further understand the mechanisms underpinning its development so as to devise novel and effective therapeutics.

Adenosine deaminases acting on RNA (ADAR) are the enzymes that convert adenosine residues into inosine (A-to-I RNA editing) in double-stranded RNAs [15]. This generates changes in the structural and functional characteristics of both the RNA itself and the translated protein, causing significant functional consequences. The mammalian ADAR gene family contains three members: ADAR1, ADAR2, and ADAR3 [16,17]. ADAR1 and ADAR2 exhibit deaminase activity but ADAR3 does not seem to have an enzymatic function [15]. Thus far, ADAR1 has two known isoforms, a shorter and constitutively active ADAR1p110a, and a full-length interferon-inducible ADAR1p150. ADAR1p150 predominantly occupies the cytoplasm, while ADAR1p110 is restricted to the nucleus, suggesting that these two isoforms exhibit distinct functions [18].

In addition to the above-mentioned editing function, i.e., A-to-I sequence change, ADAR can also fulfill its duties via an editing-independent mechanism. ADAR2, either catalytically active or inactive, was shown to mediate mir-376a2 processing by interfering with the processing of Drosha, suggesting that ADAR exhibits two distinct functions: RNA editing and RNA binding [19]. In addition, a physical interaction between ADAR1 and Dicer, thus forming a functional complex, has also been reported [20]. In addition, in metastatic melanoma, ADAR1 was shown to mediate Dicer expression through let-7 [21], and in MKN-45 (a gastric cancer cell), ADAR1 promoted miRNA processing [22]. This ADAR1/Dicer heterodimer elevates the maximum pre-miRNA cleavage rate and accelerates the loading of miRNA onto the RISC complex, subsequently enhancing the silencing of the target gene.

The major known function of ADAR1 is to mediate the immune response of cells to viral infection. It can either promote or diminish viral replication in a virus-dependent manner [23], as evidenced by the findings that ADAR1 is increased in response to various stresses, and exhibits either pro-inflammatory or anti-inflammatory activity, depending on the specific type of insult [24,25,26]. However, little is known about the involvement of ADAR1 in the pathology of viral myocarditis and the subsequent inflammation. We hypothesized that ADAR1 might be critical in the VMC pathology. In this study, we assessed the expression of ADAR1 in a mouse VMC model and used neonatal rat cardiomyocytes (NRC) and H9c2 cells as an in vitro model system to explore the implication of ADAR1 in the pathology of VMC.

## 2. Results

### 2.1. Verification of the VMC Mouse Model

BALB/c mice were injected with CVB3 or with MEM Eagle as a control. CVB3-infected mice developed symptoms such as irritability, infirmity, depression, and were less activity. On day 3 after infection, a few scattered small foci in the cardiac tissues of CVB3-treated mice were noted, indicating the presence of myocyte necrosis. On day 7 after infection, CVB3-infected mice exhibited extensive myocardial necrosis and inflammatory cell infiltration, including many lymphocytes and macrophages in and around the necrotic foci. In contrast, inflammatory cells and necrosis were barely seen in the cardiac tissues of control mice (Figure 1).

### 2.2. ADAR1 Expression in the Mouse Model of VMC and Cardiac Cell Lines Infected with CBV3

We next assessed by immunohistochemistry and Western blot the expression levels of ADAR1 in the hearts from mice from the control and VMC groups. As shown in Figure 2A, VMC mouse hearts showed increased expression of ADAR1 compared with the control hearts, and the expression was localized in the cytoplasm.

We found that the expression of ADAR1p150 was elevated in cardiomyocytes from the VMC mice, while ADAR1p110 was not significantly changed (Figure 2B). Similar results were found in neonatal rat cardiomyocytes (NRC), H9c2 cells, and cardiac fibroblasts (CF) that had beeninfected with CBV3 for 48 h (Figure 2C).

### 2.3. Interaction between ADAR1 and Dicer in the Hearts fromVMC Mice and in CBV3-Infected H9c2 Cells

Coimmunoprecipitation between ADAR1 and Dicer was performed to determine whether ADAR1 protein contributes to the regulation of Dicer (Figure 3). Our data suggest that ADAR1 tightly bound to Dicer in the VMC mouse hearts and H9c2cells infected with CBV3, suggesting that the interaction was direct.

### 2.4. Increased Level of miRNA-222 in the Hearts of VMC Mice and in CBV3-Infected H9c2 Cells

To explore the relations between miRNAs and the interaction of ADAR1 with Dicer, we used RT-qPCR to detect changes in microRNA-221, -222, -17, -151, and -432, which are related with the progress of virus infection and heart disease [27,28,29]. Compared with the control group, we found that the level of miRNA-222 was significantly higher; the others did not achieve statistical significance (Figure 4A). Based on changes in the miRNAs, we selected miRNA-222 and explored its role in NRC and CF. Interestingly, we found that miRNA-222 was also significantly elevated after infection with CVB3 compared with the control group (Figure 4B).

### 2.5. Effects of ADAR1p150 on miRNA-222 Synthesis in Cultured Cells

The single most striking observation to emerge from the data comparison was that the levels of ADAR1p150 andmiR-222 were upregulated in VMC. Interestingly, the next question was whether the regulation of miRNA-222 was related to interactions between ADAR1p150and miR-222. To further demonstrate the effects of ADAR1p150on miR-222 synthesis in cultured cells, we knocked in the gene of ADAR1p150 in H9c2 cells and CFs as depicted in Figure 5A. The upregulation in the protein level of ADAR1p150 indicated the successful knock-in of the ADAR1p150 gene (Figure 5B). We observed that the miRNA-222 expression level was elevated by approximately 200% in H9c2 cells and CFs (Figure 5B). However, when ADAR1p150 was knocked down, the results of miRNA-222 were reduced by 60–70% (Figure 5C). The above results indicate that ADAR1p150 could promote the expression of miRNA-222.

Together, these findings suggest that ADAR1p150 has an effect on miRNA-222 synthesis in cardiac cell lines.

### 2.6. miR-222 Downregulation ofPTEN Expression

PTEN was shown as a miR-222 mediator in the regulation of cell survival, migration, proliferation, and apoptosis [30,31]. In the present study, we examined the expression level of PTEN protein, which gradually decreased in H9c2 cells and CFs when ADAR1p150 was upregulated (Figure 5B). When ADAR1p150 was knocked down, the expression level of PTEN protein was increased (Figure 5C).

To evaluate the ability of miR-222 to regulate the expression level of endogenous PTEN protein in cardiac myocytes, we used TargetScan to predict the putative targets [32]. The miR-222 putative binding sites were found in the 3′-UTR of PTEN at 1269–1275 bp (Figure 6A). Then, we assessed the impact of miR-222 on PTEN expression in H9c2 cells. RT-qPCR and Western blot analysis demonstrated that miR-222 suppressed the protein levels of PTEN by approximately 40% in H9c2 cells compared to the control (Figure 6B,C). Correspondingly, mis-RNA-222 (a negative control) did not influence PTEN expression, but miR-222 inhibitor increased the PTEN level (Figure 6B,C). Taken together, miR-222 probably regulates cardiomyocyte apoptosis by suppressing PTEN expression, confirming the previous reports.

### 2.7. Regulation of PTEN by miRNA-222 Triggered by ADAR1p150 in CVB3-Induced Myocarditis

Next, we want to discuss whether the regulation process of ADAR1p150 to PTEN is dependent on miRNA-222. When the H9c2 cell line was infected with CVB3 (the ADAR1p150 was upregulated), PTEN expression was downregulated (Figure 7). Moreover, co-transfection of the miRNA-222 inhibitor almost eliminated the effect of ADAR1p150 (Figure 7). This demonstrates that ADAR1p150 specifically regulates the interaction between miRNA-222 and Dicer, and changes the expression of the PTEN protein in VMC.

### 2.8. ADAR1p150 Plays an Important Role in Maintaining Cell Viability by Regulating PTEN Expression

PTEN has been shown to be an apoptosis-associated protein [33]. Given the strong correlation of ADAR1p150 with PTEN through regulating miR-222 in VMC, we investigatedADAR1p150 in cell viability of NRC infected with CVB3. We knocked down ADAR1p150 in NRC infected with CVB3. We performed CCK-8 to check the effect of ADAR1p150 on the cell viability. Compared with the control group, the value of CCK-8 was reduced significantly after 24 and 48 h (*p* < 0.05, Figure 8A). The expression of PTEN was evaluated (*p* < 0.05, Figure 8B). In addition, we found that the expression of the pro-apoptotic protein BAX was increased and the expression of the anti-apoptotic protein Bcl-2 was reduced compared with the control group (Figure 8B, *p* < 0.05). Our data indicate that ADAR1p150 may play a crucial role in anti-apoptosis in the pathogenesis of viral myocarditis.

## 3. Discussion

VMC is a primary cause of cardiomyopathy in young adults and is linked to arrhythmia, heart failure, and even sudden death [34]. Several supportive treatments, including anti-virus, anti-oxidation therapies, and immune suppression, have been used to reverse or even block the hyperactive myocardial inflammation caused by viral infection; however, in the clinic, these approaches have not been efficient enough to improve patient survival [34,35,36]. Hence, VMC remains challenging with regard to the efficacy of treatments, and further elucidation of the fundamental mechanisms leading to VMC is essential [37,38,39].

MicroRNAs (miRNAs) mediate a diversity of cellular activities through the repression of target gene expression. Drosha and Dicer are integral parts of the miRNA functional machinery [40]. Drosha, forming a complex with DGCR8 (a dsRNA-binding protein) in the nucleus, cleaves primary transcripts of miRNAs (pri-miRNAs) to generate pre-miRNAs. Pre-miRNAs are then exported into the cytoplasm and cleaved by Dicer, thereby generating mature 21–24nt mature miRNAs [40]. Many studies have revealed a crucial role of miRNAs in the pathogenesis of VMC [41,42]. ADAR1 is involved in a variety of pathological conditions including inflammation [43,44], host defense against viral infections [45], and cancer [20,46,47]. Recent studies suggest that ADAR1 possesses two capabilities, one as an RNA editing enzyme, which involves A-to-I RNA editing that regulates RNA metabolism [20,48,49,50,51,52], and the other as an RNAi machinery component by complexing with Dicer [20].

The results of our study showed that ADAR1p150 expression in VMC was elevated, and the major part of immunostaining was localized in the cytoplasm. Similar results were found in cardiac cell lines infected with CVB3. Dicer, an endoribonuclease of the RNase III family, regulates the maturation of most miRNAs and, thus, has a significant role in numerous biological events. The evidence that ADAR/Dicer promotes miRNA processing was revealed. In our study, the elevated expression of ADAR1p150 waspredominantly localized in the cytoplasm of cardiac myocytes, the same location as Dicer, which produces mature miRNAs. Hence, we considered it reasonable to test the hypothesis that the elevated expression of ADAR1p150 has an effect on Dicer protein in the pathology of this disease. Coimmunoprecipitation between ADAR1 and Dicer was performedto determine whether ADAR1 protein contributes to the regulation of Dicer. Our data suggest that ADAR1 bound tightly to Dicer in the mouse VMC model and H9c2cells infected with CBV3, suggesting that the interaction is direct. These results verified our hypothesis.

miR-222 is involved in inflammation, apoptosis and necrosis, fibrosis, hypertrophy, and cardiac remodeling [27]. For example, knockdown of miRNA-222 suppressed neointimal lesion formation and proliferation of vascular smooth muscle cell (VSMC) after carotid angioplasty [53]. In this study, miRNA-222 was highly expressed in cardiomyocytes of VMC mouse hearts and cardiac cell lines infected with CVB3. To verify whether the highly expressed miRNA-222 was related to ADAR1p150, we knocked down the gene of ADAR1p150 in cardiac myocytes and found miRNA-222 changed correspondingly. These data indicate that ADAR1p150 mediates miRNA-222 synthesis in the pathogenesis of VMC.

We next investigated the effect of miRNA-222 on VMC. We first identified a potential binding site for mouse miR-222 and PTEN through TargetScan. PTEN is a multifunctional tumor suppressor whose major function is mediated via its lipid phosphatase activity [54,55,56]. PTEN acts as a mediator of several cellular events including apoptosis, cell survival, proliferation, and migration [57,58,59]. Recently, PTEN has been shown to mediate the function of cardiovascular and pulmonary systems [60,61]. For instance, PTEN regulates pulmonary SMC proliferation and survival, as well as cardiomyocyte hypertrophy [60,61]. Gong et al. [62] verified that the 3′-UTR of the PTEN gene contains the sequence of nucleotides complementary to the 5′ end of miRNA-222, and its activity was mediated by miRNA-222 through a reporter assay. In the present study, we used a bioinformatics tool and identified a binding site of miRNA-222 in the mouse PTEN 3′-UTR. We further demonstrated that miR-222 suppressed the expression of PTEN in cardiomyocytes. Taken together, we argue that the elevated miRNA-222 levels observed in the hearts of VMC mice are likely to contribute to the VMC pathogenesis.

In cardiac myocytes, the pro-apoptotic effect of PTEN has been proven [63]. We found the expression level of PTEN protein gradually decreased when ADAR1p150 was upregulated in cardiac myocytes. In contrast, when ADAR1p150 was knocked down, the expression level of PTEN protein was increased. The PTEN expression was downregulated when H9c2 cells were infected with CVB3 and when we co-transfected the miRNA-222 inhibitor, the effect of ADAR1p150 almost disappeared. To validate the role of PTEN in the anti-apoptotic function of ADAR1p150, we knocked down ADAR1p150 in the NRC infected with CVB3.The value of CCK-8 was reduced in the group in which ADAR1p150 was reduced significantly in the presence of CVB3 at 24 and 48 h after transfection. The protein level of PTEN was evaluated, and BAX and Bcl-2 changed correspondingly. These findings support the notion that PTEN plays essential roles in cell survival, proliferation, and apoptosis. These observations are also in line with a recent finding that ADAR may change the miRNA’s function via an editing-independent mechanism.

## 4. Materials and methods

### 4.1. Animals

All animal experiments were performed in compliance with the Guide for the Care and Use of Medical Laboratory Animals (Ministry of Health, China, 1998), and approved by the Institutional Animal Care and Use Committee at Soochow University (ECSU-201800094; 10 November 2017).

### 4.2. VMC Mouse Model

CVB3 viral stock was obtained as detailed previously [64]. A total of 30 BALB/c mice (male, 4–6 weeks old, 16–20 g) were intraperitoneally (i.p.) injected with 5 × 10^4^ plaque-forming units (PFU) of purified CVB3 as described previously [65]. Ten mice were i.p. injected with MEM Eagle as the control. Seven days post infection/injection, mice were sacrificed by pentobarbital injection (40 mg/kg, i.p.) and the hearts were excised for biochemical and histological experiments. Heart tissue that contained the left and right ventricles were fixed in 4% paraformaldehyde (pH 7.2) and paraffin-embedded for histological analysis. The remaining cardiac tissues were snap-frozen in liquid nitrogen for subsequent studies.

### 4.3. Histology and Immunohistochemistry

Murine tissue was stained as described elsewhere [66]. Tissue sections were prepared at 4-µm thickness. Immunohistochemical staining for ADAR1 (Santa Cruz Biotechnology, Dallas, TX, USA) was performed as previously described [66]. All slides were counterstained with hematoxylin. Slides were viewed with a Zeiss Axioskop 40 microscope (Carl Zeiss, Jena, Germany).

### 4.4. Culture of Cardiomyocytes and Fibroblasts

Cardiomyocytes were prepared through trypsin digestion from neonatal mice within 72 h of birth, as previously detailed [67]. Cardiac fibroblasts and cardiomyocytes were separated through pre-plating. Cardiomyocytes isolated by this method were cultured in collagen-pre-coated culture wells, and 95% exhibited typical cardiomyocyte structure. Isolated cardiac fibroblasts were cultured in DMDM supplemented with 10% FBS.

### 4.5. H9c2 Cell Culture

H9c2 cells, a cardiac cell line (CRL146, ATCC), were grown at a density of approximately 10^5^ cells/cm^2^ and cultured as monolayers in DMEM supplemented with 10% FBS, nonessential amino acids (1%), streptomycin (100 μg/mL) and penicillin (100 IU), and glutamine (2 mM), in a 37 °C incubator with 5% CO_2_ and water vapor. The medium was replenished every 2 days.

### 4.6. Western Blots

Cardiac tissues and transfected cells were homogenized in RIPA buffer (Heart, Beijing, China). BCA Protein Assay Kit was used to determine the protein concentrations. Then, 50 μg of total protein per sample was subjected to 12% sodium dodecyl sulfate–polyacrylamide gel electrophoresis (SDS-PAGE), followed by transfer to a PVDF membrane (BioRad, Hercules, CA, USA). The membrane was then incubated with the primary antibody of interest, washed with PBS-T, and incubated with an appropriate secondary antibody. Protein bands were quantified with ImageQuant software. The β-actin level was used as an internal control. The following primary antibodies were used: ADAR1 (Santa Cruz Biotechnology, Dallas, TX, USA), Dicer (Biorbyt, Cambridge, UK), PTEN (Abcam, Cambridge, UK), BAX (Cell Signaling Technology, Danvers, MA, USA); Bcl-2 (Bioworld, Nanjing, China); β-actin (Cmctag, Shanghai, China); GAPDH (Cmctag, Shanghai, China). Three independent experiments were performed for each sample.

### 4.7. Coimmunoprecipitation (Co-IP)

Whole cell lysates were prepared in the lysis buffer (10 mM Tris-HCl, pH7.5, 150 mM NaCl, 1% Triton X-100, 25 μg/mL leupeptin, and 25 μg/mL aprotinin). Briefly, cells were lysed on ice for 20 min and centrifugated at 13,000× *g* for 20 min, followed by supernatant collection. The anti-ADAR1 antibody or anti-Dicer antibody was used to bind to the desired protein complex. Immunocomplex or MAG–Fc was precipitated with protein A-conjugated beads (Amersham Pharmacia Biotech, Little Chalfont, UK). The precipitates were subjected to Western blot analysis.

### 4.8. Upregulation and Knockdownof ADAR1p150 Protein

ADAR1p150 knockdown and up-expression vectors, as well as a negative control (NC), were purchased from GeneChem (Shanghai, China). The plasmid was delivered into the cells by the lipid carrier lip3000 (Thermo Fisher Scientific, Waltham, MA, USA). The small interfering RNA sequence for ADAR1p150 was 5′-AGGGACATGGGCTATGGGAAT-3′. Transfection results were confirmed by Western blot analyses after 48 h.

### 4.9. miRNA Synthesis and miRNA Inhibitor

Murine miR-222 and its mutated RNA oligos were synthesized by RiboBio (Guangzhou, China). The sequences of miR-222 mimic and inhibitor were as follows (5′–3′): miR-222 mimic, AGCUACAUCUGGCUACUGGGU; miR-222 inhibitor, the sequence of the negative control (NC) is confidential. We transfected the cells by using lipo3000 (Thermo Fisher Scientific, Waltham, MA USA) for 48 h.

### 4.10. Transfection of miRNAs

miRNA mimic, miRNA inhibitor, or siRNAs (Ribobio, Guangzhou, China) were transfected into cells seeded in 6-well dishes using lipo3000 (Thermo Fisher Scientific). After overnight (12 h) culture, cells were placed in fresh culture medium for an additional 6 or 24 h culture.

### 4.11. RNA Purification andReverse Transcription Polymerase Chain Reaction (RT-PCR)

Trizol was used to purify total RNA from heart tissue and cardiac cells according to the protocol provided by the manufacturer. After denaturing RNA at 95 °C for 5 min, RT-PCR was performed in a total volume of 20 μL with the following protocol: 70 °C for 5 min, 37 °C for 5 min, 42 °C for 60 min, and 70 °C for 10 min. After RT-PCR, 40 cycles of the PCR protocol (95 °C (10 s) and 60 °C (40 s)) wasfollowed in an ABI StepOne Plus Sequence Detection System (Applied Biosystems, Waltham, MA, USA). GAPDH or U6 was used as an internal control.

### 4.12. Analysis of Cardiomyocyte Viability

The cell counting kit-8 (CCK-8; Dojindo, Shanghai, China) was performed to measure the viability of cardiomyocytes. Briefly, equal amounts of cells were cultured in FBS (5%)-l-DMEM in 96-wellplates for 24 h and then treated with CVB3 for 24 or 48 h after ADAR1p150 was knocked down. According to the supplier’s instructions, 10 µL/well of CCK-8 was added to each well for a further 3 h incubation at 37 °C. The ELx808 Absorbance Microplate Reader with an optical density (OD) at 450 nm was used to calculate cell viability.

### 4.13. Statistical Analysis

All data are presented as the mean ± standard error of the mean (SEM). The two-tailed independent Student’s *t*-test and ANOVA were used for the statistical analysis. *p* < 0.05 was considered statistically significant. All statistical analyses were performed using SPSS, version 12.0 (SPSS Inc., Chicago, IL, USA).

## 5. Conclusions

In summary, we have demonstrated that ADAR1p150 promotes miRNA-222 through the formation of a complex with Dicer and, subsequently, that miRNA-222 regulates the expression of PTEN in VMC. ADAR1p150 plays an important role in maintaining cell viability. Our results provide novel insight into the mechanisms of viral myocarditis and point out that ADAR1 plays a key role in cardiac myocytes. By understanding the molecular basis underlying ADAR1p150 function, novel information might be offered. Such informationmay promote new therapeutics for VMC in the future. However, more research will be needed to understand how the editing-dependent and -independent mechanisms cooperate to alter the expression level and function of specific miRNAs and their target genes.

## Figures and Tables

**Figure 1 ijms-20-00407-f001:**
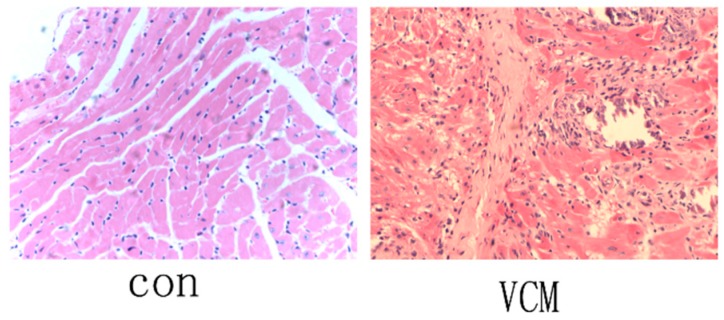
H&E stain of the left ventricle of mice from the control and viral myocarditis (VMC) groups (original magnification 9100). Many macrophages and lymphocytes had infiltrated the myocardium on day 7 after infection.

**Figure 2 ijms-20-00407-f002:**
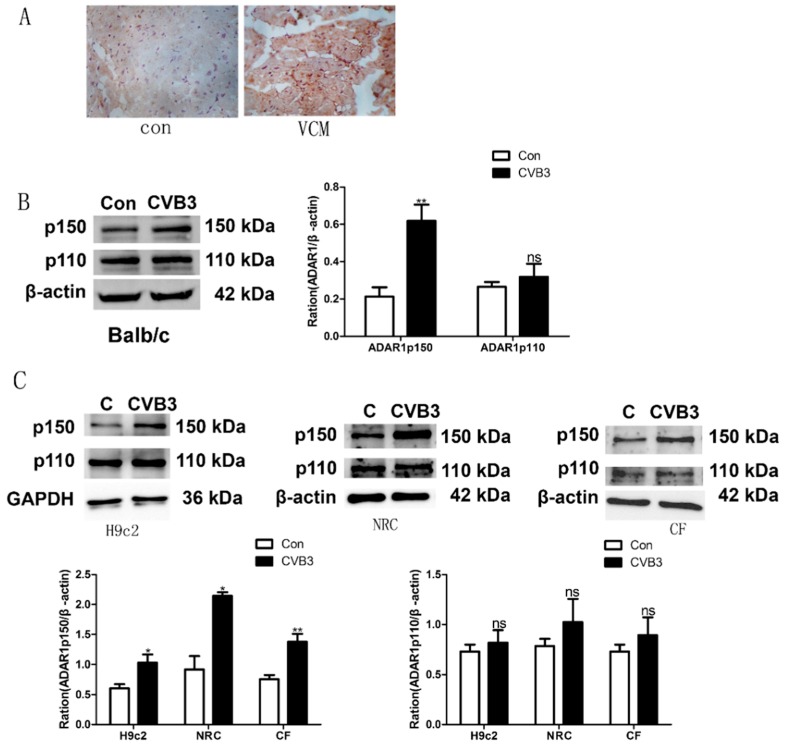
Adenosine deaminases acting on RNA 1 (ADAR1) expression in the hearts of VMC mice and CBV3-infected H9c2cells. (**A**) Immunohistochemical staining revealed that ADAR1 occupied the cytoplasm and its expression was increased in the cardiomyocytes from the VMC mice (original magnification 200×). (**B**) The expression of ADAR1p150 was increased in VMC (BALB/c mice) on the seventh day after infection, while ADAR1p110 was not. (**C**) The same method was used to analyze changes inADAR1 in the H9c2 cell line, primary cardiac myocytes (NRCs), and cardiac fibroblasts (CFs), respectively,48 h after CVB3 infection; the result is consistent with heart tissue. * *p* < 0.05, ** *p* < 0.01.

**Figure 3 ijms-20-00407-f003:**
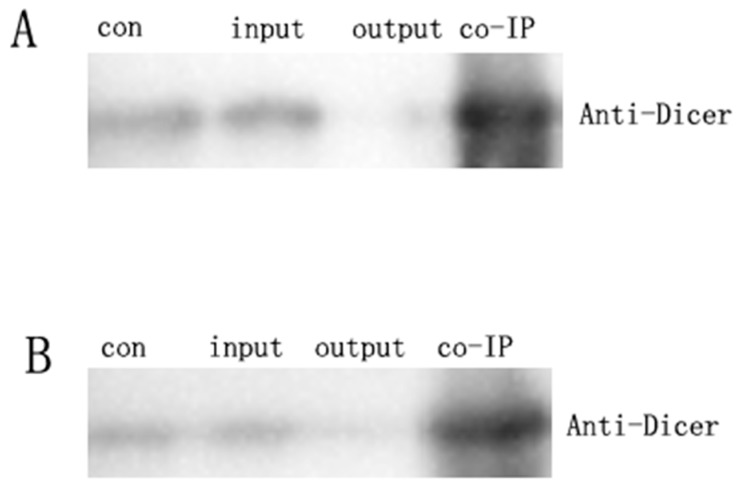
(**A**)ADAR1p150 promotes Dicer complex formation. ADAR1p150 interacts with Dicer in the VMC mouse model. (**B**) ADAR1p150 promotes Dicer complex formation. ADAR1p150 interacts with Dicer in the CBV3-infected H9c2 cells. Coimmunoprecipitation analysis was performed with the indicated antibodies. The experiment was conducted three times.

**Figure 4 ijms-20-00407-f004:**
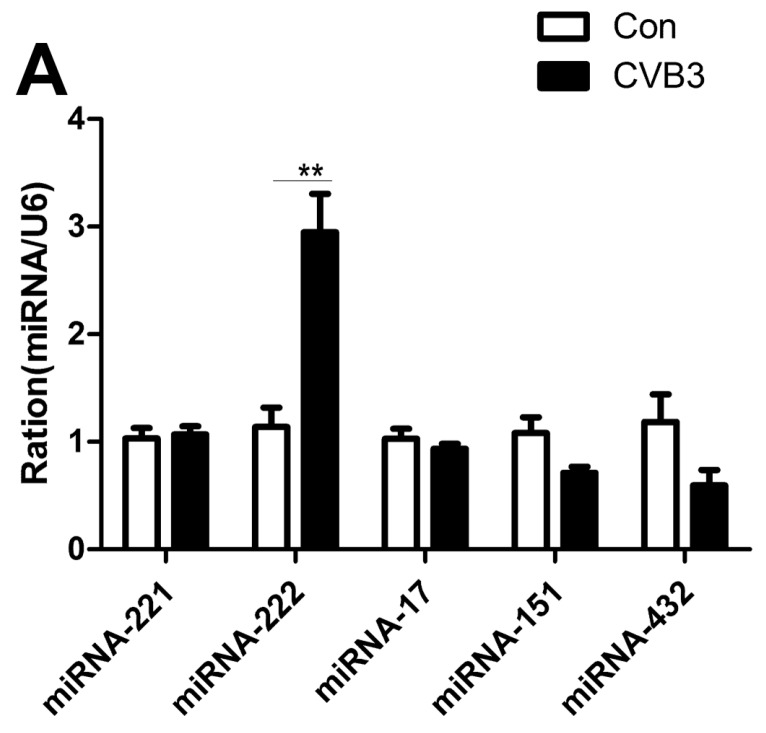
Increased level of miRNA-222 in VMC in the mouse model of VMC and cardiac cell lines infected with CBV3. (**A**) RT-qPCR was used to detect changes inmiRNA-221, -222, -17, -151, and -432, respectively, in myocardial tissue. (**B**) The miRNA-222 of relative quantification was further determined in primary cardiac myocytes and cardiac fibroblasts. Data represent the mean ± SEM from the control (Con) and CVB3-infected groups, ** *p* < 0.01.

**Figure 5 ijms-20-00407-f005:**
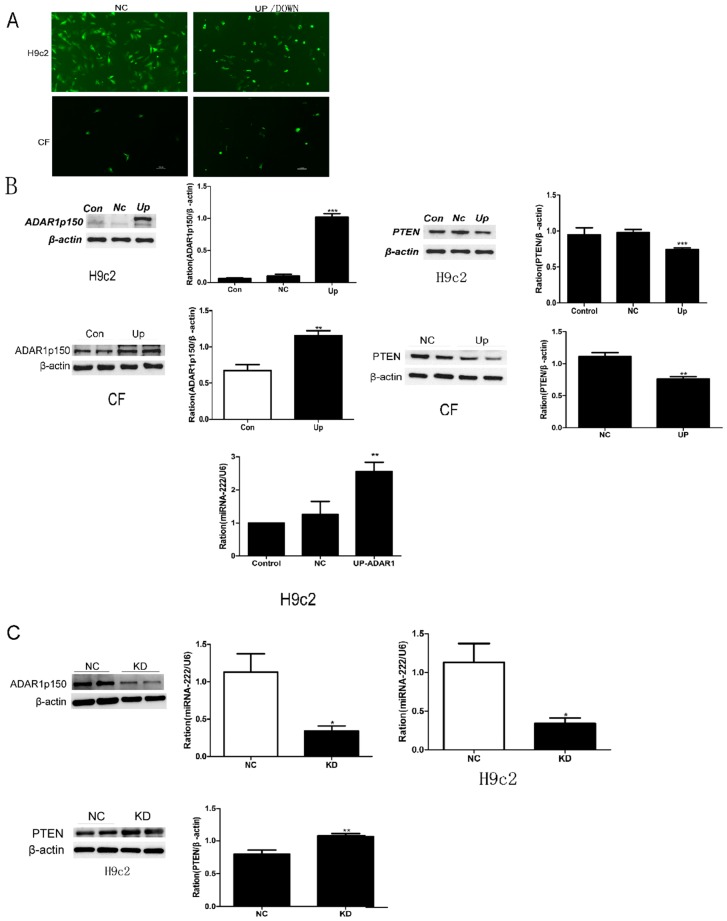
Effects of ADAR1p150 on miRNA-222 synthesis in cultured cells and regulation of phosphatase-and-tensin (PTEN) expression by miRNA-222. (**A**) GFP as a marker protein was detected by immunofluorescence after 48 h transfection in the H9c2 cell line and CFs (cardiac fibroblasts). As shown in the picture, the transduction efficiency was always over 80%. (**B**) After confirming that ADAR1p150 high expression transfection was successful, miRNA-222 and PTEN were quantitatively or relatively quantified. (**C**) After inhibiting the expression of ADAR1P150, miRNA-222 and PTEN were quantitatively or relatively quantified. Data represent the mean ± SEM from the control (Con)and infected groups, negative control (NC)knocked down (KD), * *p* < 0.05, ** *p* < 0.01, *** *p* < 0.001.

**Figure 6 ijms-20-00407-f006:**
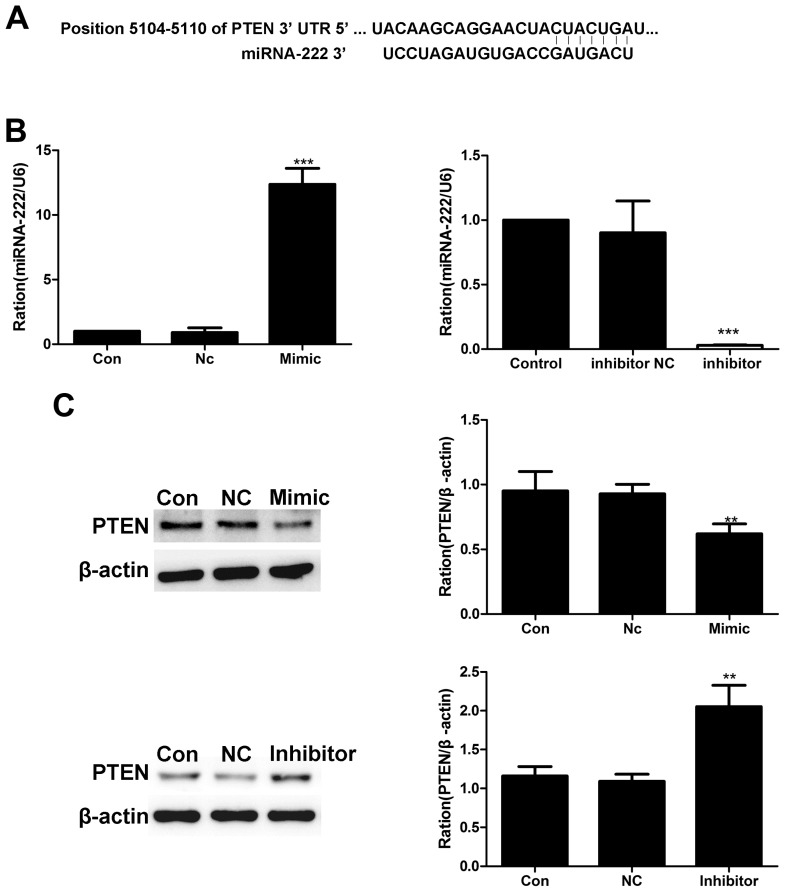
PTEN is a target protein of miRNA-222. (**A**) According to the principle of base pairing to deduce the binding of miR-222 to the predicted sequence in the 3′-UTR of mouse PTEN mRNAs, seven complementary sites were demonstrated. (**B**) We successfully transferred the mimic and inhibition of miRNA-222 into H9c2 cells, as determined by quantitative-PCR. (**C**) The change in PTEN protein in H9c2 cells when miRNA-222 was upregulated and downregulated, respectively. NC = negative control. Data represent the mean ± SEM from the Con and infected groups, ** *p* < 0.01, *** *p* < 0.001.

**Figure 7 ijms-20-00407-f007:**
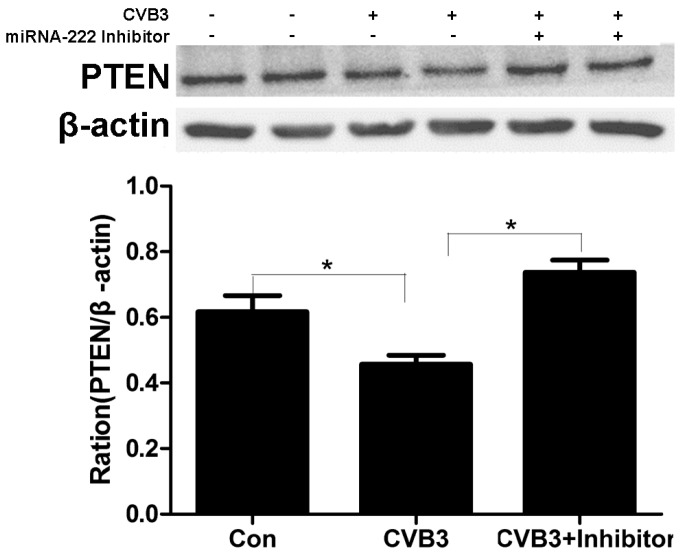
PTEN regulation by miRNA-222 in hearts of VMC mice. The PTEN expression was downregulated when H9c2 cells were infected with CVB3(* *p* < 0.05). When we co-transfected the miRNA-222 inhibitor, the effect of ADAR1p150 almost disappeared.

**Figure 8 ijms-20-00407-f008:**
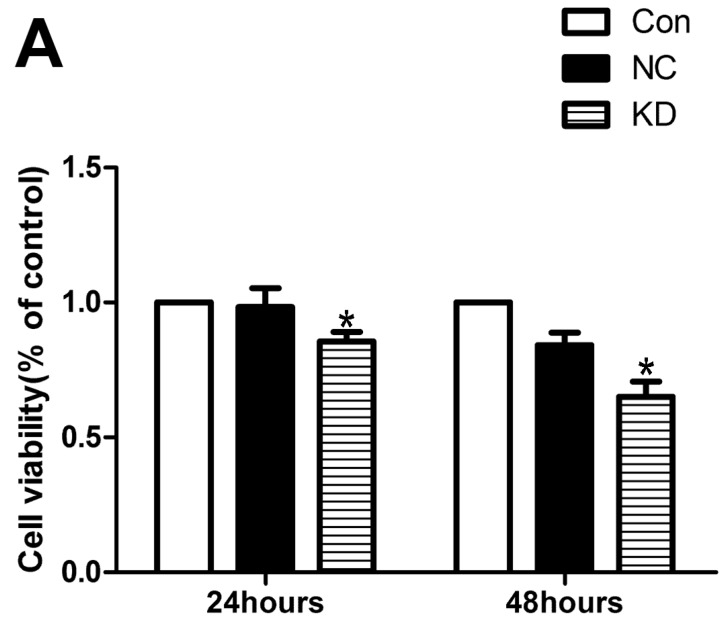
The effect of ADAR1p150 on cell viability in NRC infected with CVB3. (**A**) The value of CCK-8was significantly reduced after we knocked down ADAR1p150 in the NRC infected with CVB3 at 24 and 48h (* *p* < 0.05). (**B**) The expression of PTEN and BAX were evaluated (* *p* < 0.05, ** *p* < 0.01) and Bcl-2 was reduced compared with the control group (* *p* < 0.05).

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
