# Peer review of "ADAR1p150 Forms a Complex with Dicer to Promote miRNA-222 Activity and Regulate PTEN Expression in CVB3-Induced Viral Myocarditis"

_ijms, 2019, doi:10.3390/ijms20020407_

Reviewer 1 Report

VMC is a primary cause of cardiomyopathy in young adults and is linked to arrhythmia, heart failure, and even sudden death. Several supportive treatments, including anti-virus, anti-oxidation therapies, and immune suppression, have been used to reverse or even block the hyperactive myocardial inflammation caused by viral infection; This work demonstrated that ADAR1p150 promotes miRNA-222 through forming to 263 complex with Dicer, and subsequently, miRNA-222 regulates the expression of PTEN in VMC. It is a very solid work and it should be published without corrections given the Underlying the molecular basis underlying ADAR1p150 function may offer novel 267 information, which may promote new therapies for VMC in the future.

Author Response

Ms. No.: ijms-418462

International journal of molecular sciences

Dear Editor,

Thank you for your kind comments on our manuscript.

Reviewer 2 Report

The manuscript entitled “ADAR1p150 forms a Complex with Dicer to Promote miRNA-222 Activity and Regulate PTEN Expression in Viral Myocarditis” investigated the mechanistic role of ADAR1p150 in a mouse model of Coxsackievirus B3 (CVB3)-induced viral myocarditis.  The study demonstrated a regulatory function of the ADAR1p150/dicer on the PTEN expression via upregulating miR-222, which was linked to anti-apoptotic signals of cardiomyocytes.

The topic is of translational value and appeals to a multidisciplinary readership of researchers.  The study design and methodology are appropriate, the findings are presented clearly and add to the literature regarding the potential role of RNA editing enzymes in the pathogenesis of viral myocarditis.

Major comments:

1. In the study by Corsten et al (Eur Heart J. 2015 Nov 7;36(42):2909-19), the miR-221/222 cluster was identified as a protective factor in myocarditis, and up-regulated by CVB3 infection in neonatal rat cardiomyocytes. However, in the present study, only miR-222 was elevated in the myocardial tissue infected with CVB3 (Figure 4A).  Could the authors expand the discussion and comment on this?

2. Expression of Bax and Bcl-2 was used when determining the pro- or anti-apoptotic effect associated with ADAR1p150.  Could the authors provide data of other pro-apoptotic and anti-apoptotic proteins?

Minor comment:

1. The experiment is conducted in Coxsackievirus B3 (CVB3)-induced viral myocarditis.  It is suggested to reflect this in the title of manuscript.

Author Response

Ms. No.: ijms-418462

International journal of molecular sciences

Dear Editor,

Thank you for your kind comments on our manuscript (Ms. No.: ijms-418462) entitled “ADAR1p150 forms a complex with Dicer to promote miRNA-222 activity and regulate PTEN expression in viral myocarditis”. We have carefully revised the manuscript according to the reviewers’ comments. Based on the suggestions, we have made an extensive modification on the original manuscript. Detailed revision was shown as follows.

1. Comment:  In the study by Corsten et al (Eur Heart J. 2015 Nov 7;36(42):2909-19), the miR-221/222 cluster was identified as a protective factor in myocarditis, and up-regulated by CVB3 infection in neonatal rat cardiomyocytes. However, in the present study, only miR-222 was elevated in the myocardial tissue infected with CVB3 (Figure 4A).  Could the authors expand the discussion and comment on this?

Response: In this study, miRNA-222 was highly expressed in cardiomyocytes of VMC mouse hearts, while the expression of miRNA-221 did not achieve the statistical significance. We made the model of VMC with BALB/c mice, while Corsten et al with C3H and C57Bl6N mice. We consider maybe the difference type of the mice infected with CVB3 lead to the different expression of miRNA-221.

2. Comment:  Expression of Bax and Bcl-2 was used when determining the pro- or anti-apoptotic effect associated with ADAR1p150.  Could the authors provide data of other pro-apoptotic and anti-apoptotic proteins?

Response: In cardiac myocytes, the pro-apoptotic effect of PTEN has been proved. Bcl-2 and BAX are the representative protein with anti-apoptotic and pro-apoptotic function. So we measured the expression level of BAX and bcl-2, and they changed correspondingly support the notion that PTEN plays essential roles in cell survival, proliferation, and apoptosis.

3. Comment:  The experiment is conducted in Coxsackievirus B3 (CVB3)-induced viral myocarditis.  It is suggested to reflect this in the title of manuscript.

Response: Accorrding to the reviewer’s recommendation, the title was changed to “ADAR1p150 forms a complex with Dicer to promote miRNA-222 activity and regulate PTEN expression in CVB3-induced viral myocarditis” .